# Assessing Versatile Machine Learning Models for Glioma Radiogenomic Studies across Hospitals

**DOI:** 10.3390/cancers13143611

**Published:** 2021-07-19

**Authors:** Risa K. Kawaguchi, Masamichi Takahashi, Mototaka Miyake, Manabu Kinoshita, Satoshi Takahashi, Koichi Ichimura, Ryuji Hamamoto, Yoshitaka Narita, Jun Sese

**Affiliations:** 1Artificial Intelligence Research Center, National Institute of Advanced Industrial Science and Technology, 2-3-26 Aomi, Koto-ku, Tokyo 135-0064, Japan; rkawaguc@cshl.edu; 2Division of Molecular Modification and Cancer Biology, National Cancer Center Research Institute, 5-1-1 Tsukiji, Chuo-ku, Tokyo 104-0045, Japan; satoshi.takahashi.fy@riken.jp (S.T.); rhamamot@ncc.go.jp (R.H.); sesejun@humanome.jp (J.S.); 3Cold Spring Harbor Laboratory, 1 Bungtown Rd, Cold Spring Harbor, NY 11724, USA; 4Department of Neurosurgery and Neuro-Oncology, National Cancer Center Hospital, 5-1-1 Tsukiji, Chuo-ku, Tokyo 104-0045, Japan; yonarita@ncc.go.jp; 5Department of Diagnostic Radiology, National Cancer Center Hospital, 5-1-1 Tsukiji, Chuo-ku, Tokyo 104-0045, Japan; mmiyake@ncc.go.jp; 6Department of Neurosurgery, Osaka University Graduate School of Medicine, 2-2 Yamadaoka, Suita 565-0871, Japan; mail@manabukinoshita.com; 7Cancer Translational Research Team, RIKEN Center for Advanced Intelligence Project, 1-4-1 Nihonba-shi, Chuo-ku, Tokyo 103-0027, Japan; 8Division of Brain Tumor Translational Research, National Cancer Center Research Institute, 5-1-1 Tsukiji, Chuo-ku, Tokyo 104-0045, Japan; kichimur@ncc.go.jp; 9Humanome Laboratory, 2-4-10 Tsukiji, Chuo-ku, Tokyo 104-0045, Japan

**Keywords:** glioma, machine learning, radiogenomics, IDH, MGMT

## Abstract

**Simple Summary:**

Radiogenomics enables prediction of the status and prognosis of patients using non-invasively obtained imaging data. Current machine learning (ML) methods used in radiogenomics require huge datasets, which involve the handling of large heterogeneous datasets from multiple cohorts/hospitals. In this study, two different glioma datasets were used to test various ML and image pre-processing methods to confirm whether the models trained on one dataset are universally applicable to other datasets. Our result suggested that the ML method that yielded the highest accuracy in a single dataset was likely to be overfitted. We demonstrated that implementation of standardization and dimension reduction procedures prior to classification, enabled the development of ML methods that are less affected by the multiple cohort difference. We advocate using caution in interpreting the results of radiogenomic studies of the training and testing datasets that are small or mixed, with a view to implementing practical ML methods in radiogenomics.

**Abstract:**

Radiogenomics use non-invasively obtained imaging data, such as magnetic resonance imaging (MRI), to predict critical biomarkers of patients. Developing an accurate machine learning (ML) technique for MRI requires data from hundreds of patients, which cannot be gathered from any single local hospital. Hence, a model universally applicable to multiple cohorts/hospitals is required. We applied various ML and image pre-processing procedures on a glioma dataset from The Cancer Image Archive (TCIA, *n* = 159). The models that showed a high level of accuracy in predicting glioblastoma or WHO Grade II and III glioma using the TCIA dataset, were then tested for the data from the National Cancer Center Hospital, Japan (NCC, *n* = 166) whether they could maintain similar levels of high accuracy. Results: we confirmed that our ML procedure achieved a level of accuracy (AUROC = 0.904) comparable to that shown previously by the deep-learning methods using TCIA. However, when we directly applied the model to the NCC dataset, its AUROC dropped to 0.383. Introduction of standardization and dimension reduction procedures before classification without re-training improved the prediction accuracy obtained using NCC (0.804) without a loss in prediction accuracy for the TCIA dataset. Furthermore, we confirmed the same tendency in a model for IDH1/2 mutation prediction with standardization and application of dimension reduction that was also applicable to multiple hospitals. Our results demonstrated that overfitting may occur when an ML method providing the highest accuracy in a small training dataset is used for different heterogeneous data sets, and suggested a promising process for developing an ML method applicable to multiple cohorts.

## 1. Introduction

Magnetic resonance imaging (MRI) is widely used for cancer diagnoses. It is most frequently used to diagnose the pathology of brain tumors [1,2]. Besides conventional diagnostic information, MRI data may also contain phenotypic features of brain tumors, which are potentially associated with the underlying biology of both the tumor and the patient [3,4]. Thus, MRI is drawing attention as a source of information that may be utilized to predict genomic or clinical backgrounds of brain tumor patients, leading to the development of treatment strategies [2]. This field of research is termed “radiogenomics” [5].

Glioma presents a predominant target for radiogenomics, because identification of biomarkers that may help improve glioma patient outcomes is considered to be an urgent task [2,6,7]. WHO Grade IV glioblastoma (GBM), in particular, results in distinctly severe outcomes, compared to other WHO Grade II and III gliomas (lower grade gliomas: LrGG) and the five-year survival rate of GBM is as low as 6.8%, while that of LrGG is higher such as 51.6% for diffuse astrocytoma or 82.7% for oligodendroglioma [8]. Recent reports indicate that numerous genetic mutations may play a role in the heterogeneity of gliomas. The molecular landscape has been remarkably transformed by the detection of key genetic mutations or epigenetic modifications, such as IDH1 and IDH2 mutations, the TERT promoter mutation, chr1p/19q codeletion, and O-6-Methylguanine-DNA methyltransferase (MGMT) promoter methylation [7]. IDH1 and IDH2 mutations (hereafter referred to as IDH mutation) are mutually exclusive “truncal mutations”, which are frequently associated with better prognoses for gliomas, such as Grade II and Grade III gliomas, and thereby considered as prognostic factors for gliomas [9,10]. MGMT promoter methylation status is also known to be a prognosis factor of GBM associated with temozolomide, which is used as a first-line, standard chemotherapeutic agent for GBM patients.

In addition, recently, the WHO grading system of central nervous system tumors was updated to require genetic testing including IDH mutation, TERT promoter mutation, H3K27M mutation, 1p/19q codeletion for the precise diagnosis of gliomas. Currently these amendments can offer better understanding of glioma biology, but also raise a new problem that all the tumor tissues supposed to be a glioma should be ideally taken by surgery or biopsy and analyzed for genetic testing, but not all countries have enough pathological diagnosis capacity. As such, it is clinically useful if genetic backgrounds could also be obtained at the same time when the first pre-operative images of glioma patients are captured.

Glioma is a rare cancer and, therefore, the number of patients at any given hospital is not necessarily large. Some large public databases, such as The Cancer Imaging Archive (TCIA) GBM and LGG [6] or Repository of Molecular Brain Neoplasia Data (REMBRANDT) were published [11] in order to enhance radiogenomic studies targeting glioma. Many recent studies have applied a wide variety of machine learning (ML) methods from volumetric features to cutting-edge deep learning (DL) methods, in order to predict the grade and grouping of tumors [12,13], including IDH mutation [14,15], MGMT methylation status [16,17,18,19,20,21], survival [22,23,24,25,26], as well as other combinations of patient backgrounds [27,28,29,30,31,32,33,34]. On the other hand, studies using local datasets showed significant diversity in prediction accuracy for the same prediction target. For example, radiogenomic features are found to predict MGMT methylation status with 83% accuracy using deep learning [28], although another study achieved the accuracy only around 62% for the same data source [35]. While numerous studies have validated the robustness of gene expression signatures for GBM [15,36], only a few have used multiple datasets from different sources [32,37]. Although, based on radiomic image features, these studies could not show consistent improvement in prediction accuracy without mixing cohorts, or using datasets where the number of patients was not sufficient to reduce the variance of classification performances obtained by cross validation. Furthermore, most studies examine the prediction accuracy of the best methods within one dataset and do not show the results from the viewpoint of the performance stability across the datasets, particularly for the difference between public anonymized dataset and local data [38]. The reproducibility of radiogenomic studies has already gathered attention for the evaluation of different preprocessing strategies and tumor annotations [39,40]. Image data that depend on the hospital-specific MRI sequence parameters or image processing for analyses tend to have many systematic and inevitable differences, called batch effects [41,42]. A spatial standardization, which is an approach to scale each brain image to a reference brain, is one promising way to reduce the impact of the systematic differences, while even the choice of standardization is impactful for the performance stability [43]. This raises a query as to whether a model developed using images obtained from a public database, which are generally pre-processed for anonymization, is applicable to images obtained from a local hospital in a practical way. This type of problem is the main subject of transfer learning in the ML domain because re-training of the entire models demands a huge resource as well as annotated training data from a local hospital [44].

The objective of the present study is to identify the important factors to establish ML models that exhibit a high level of performance, not only in a single-cohort, but also in other hospital datasets. We examined the stability of prediction accuracy using two different MRI datasets, wherein the public TCIA dataset was used to generate a model, which was then directly applied to the National Cancer Center Hospital Japan (NCC) dataset. Because the public dataset is already preprocessed, this comparison gives us the clarification whether the trained model is still effective for datasets from local hospitals beyond the various hospital or cohort-specific biases. According to our results, the best model is shown to suffer from the over-fitting problem for another dataset, especially when the model does not include either standardization or dimension reduction processes. It would be a robust and practical solution to choose the better model from multiple models trained for the public dataset using a partially annotated local dataset.

## 2. Materials and Methods

### 2.1. TCIA and NCC Cohort

This study was approved by the National Cancer Center Institutional Review Board (study number: 2013-042). The NCC cohort dataset consisted of MRI scans, clinical information, and genomic information collected from 90 GBM and 76 LrGG patients who were treated at NCC between 2004 and 2017. Patient information is summarized (Table 1 and Appendix A). MRI data contained 4 widely used data types as follows: T1-weighted imaging (T1WI); gadolinium-enhanced T1-weighted imaging (Gd-T1WI); T2-weighted imaging (T2WI); and fluid-attenuated inversion recovery (FLAIR) sequence. All images were acquired in an axial direction with thicknesses in the range of 4–6 mm.

As another cohort, the public image and background data of 102 GBM and 65 LGG patients were obtained from the training dataset of Brain Tumor Segmentation (BraTS) challenge 2018 [6], which is based on TCIA information. This public dataset is constructed for the contest of brain tumor segmentation methods and will hereafter be referred to as the TCIA dataset. The images were already co-registered, skull-stripped, and re-sampled to a size of 240 × 240 × 155. The genomic and clinical information from these were described previously [45]; (Table 1 and Appendix A). Patients with a set of at least 3 types of MRI scans, tumor annotation, and genomic information were selected.

### 2.2. Region of Interest (ROI) Extraction

Tumor regions of the NCC cohort dataset were examined by a skilled radiologist (MM), who has been practicing for 18-years, using ImageJ [46]. For all 4 MRI types, 2 categories of ROI information tumor (enhanced in Gd-T1WI) region and edema (high intensity lesion in T2WI and FLAIR) were contoured via single strokes. Image features were computed for the region inside of the outline. ROIs were also transformed via realigning, co-registering, and normalizing in the same manner. Tumor and edema ROI of the TCIA dataset were provided by BraTS. Pixels annotated as “necrosis” or “enhancing tumor region” were labeled as tumor, and regions annotated as “necrosis”, “edema”, “non-enhancing tumor”, and “enhancing tumor region” were treated as edema ROI, which corresponds to the edema region and its inside. All results for features computed for tumor ROI or without ROI are shown in Appendix A.

### 2.3. Preprocessing of Brain Images

NCC-derived MRI data was converted from digital imaging and communications in medicine (DICOM) to the Neuroimaging Informatics Technology Initiative (NIfTI) format with neurological space, using SPM12 [47]. Skull-stripping was carried out via the standard_space_roi command in FSL-BET library [48]. For the TCIA dataset, we applied normalization and brain standardization to the downloaded MRI data in the NIfTI format that was already skull-stripped and resampled after normalization and anonymization processes.

Two frequently used spatial standardization procedures were applied to adjust differences in individual brain sizes and shapes; one was Tissue Probability Maps (TPM) in MNI space using SPM12 (referred to SPM), while the other was average brain space compiled by MICCAI multi atlas challenge data using ANTs [49]. We also applied two normalization methods to regulate differences in pixel brightness. Details pertaining to an entire standardization, such as pixel normalization, mapping, and skull-stripping processes are shown in Appendix A.

### 2.4. Calculating Image and Patient Features for ML

To examine the classification efficiency of a wide variety of image features, a maximum of 16,221 features were generated for each patient, using both images and clinical records for the following ML training and prediction classifiers. Identical procedures were applied to both the NCC cohort and the TCIA dataset. The features are summarized below, where *M* represents the number of features generated in each category (see Appendix A).
Basic features (MRI scans; M = 28 × 4): statistics of pixel values from 4 types of MRI scans. Twenty-eight features were generated from each MRI type, consisting of percentile pixel values of the ROI as well as whole brain regions as a control with steps increasing by 10%. Several other statistics, such as mean, median, min, max, dimension size (x, y, z), and centroid coordinates of the whole brain were also included to control patient-specific image pattern.Pyradiomics-based features (MRI scans; M = 960 × 4): first order statistics, shapes, and textures were calculated by pyradiomics (v2.1.0), which is frequently used software for radiomic analysis [3]. Pyradiomics was used for radiomic feature extraction from 2D and 3D images with ROI information. All parameters were computed except for NGTDM due to the long running time.Pre-trained DL-based features (MRI scans; M = 3072 × 4): the DL model Inception-ResNet v2, pre-trained on ImageNet database, was used to obtain general image features. Outputs of the second to last layer were used as image features. Because the number of slices containing tumors was different for each patient, averages and sums of the outputs along the z-axis were calculated.Anatomical tumor location (MRI scans; M = 30): a vector representing occupancies of each anatomical region calculated via FSL (v5.0.10) [48] was used to represent information pertaining to 3D tumor position.Clinical information (M = 3): features of each patient, such as sex, age, and Karnofsky Performance Status (KPS) were used as additional features.

Most image features were computed only for edema (tumor, or all brain) regions, while basic features contained the metrics for both edema and whole image. In addition, the computation of DL-based features used rectangular images to cover all target tumor regions with the margin of 1 pixel using the command line tool “ExtractRegionFromImageByMask” included in the ANTs library [49]. The results shown in the main script were obtained using edema ROIs because of the high prediction performances. Those with tumor ROIs and without ROIs are found in Appendix A.

### 2.5. Feature Selection and Dimension Reduction Methods

One feature selection and two-dimension reduction methods were tested to demonstrate the efficacy of feature generation for transfer learning on radiogenomics. Logistic regression with L1 regularization and linear discriminant analysis, with the maximum number of features set to 200, was used for feature selection. Principal component analysis (PCA) and non-negative matrix factorization (NMF) were used as dimension reduction methods. PCA and NMF approximate input data into the low-dimensional matrices, where the dimension of the column or the row is a user-defined value, thereby reducing noise and integrating similar features into a single dimension. In both methods, the number of dimensions following decomposition was set to 8, 20, 40, and 200, in order to extract the primary features in the 4 MRI scan types. These analyses were implemented using Python 3 scikit-learn library.

### 2.6. Classification and Performance Evaluation for ML Classifiers

In order to evaluate the robustness of ML classifiers to dataset differences, widely-used and accurate ML classifiers were applied to the classification procedure as follows: random forest, k-nearest neighbor (with k set to 1, 3, and 5), support vector machine, linear discriminant analysis, AdaBoost, and XGBoost. For purposes of classification, feature vectors were applied without normalization to avoid effects of patient imbalance on each dataset. Five-fold cross validation was performed for the TCIA dataset in order to evaluate these methods. Next, in order to select the best-fit model for NCC, the model was applied to half of the NCC data that had been randomly sampled to validate each prediction problem (referred to as NCC validation). Lastly, the final accuracy of the model was estimated using the remaining half of the NCC data (NCC test). The area under the receiver operating characteristic curve (AUROC) represented the measure of accuracy used to evaluate the general robustness of the models.

### 2.7. Publicly Available Brain Image Analysis Toolkit (PABLO)

The scripts used in this study are available at our repository, named PABLO (https://github.com/carushi/PABLO, accessed on 14 June 2021). PABLO is a versatile platform for analyzing combinations of methods, consisting of three steps widely used in MRI analysis; (1) standardization; (2) dimension reduction; and (3) classification. Standardization projects a patient head shape to the single standard to alleviate the individuality of brain images with adjusting the distribution of pixel brightness. Dimension reduction extracts or generates important features from the data, often resulting in stable classification results.

## 3. Results

Statistics of patients in the TCIA and NCC datasets are shown in Table 1. Both datasets displayed similar tendencies in terms of the number of GBM/LrGG, mutations and epigenomic status. On the other hand, the distribution of image feature values indicated that the distributions of the two datasets were clearly separated regardless of the property of patients (Figure 1). High-dimensional image features from each patient were calculated according to standard MRI analysis methods as described in Materials and Methods, and visualized in two dimensions via t-SNE [50]. While both TCIA and NCC datasets contained information from GBM, as well as LrGG patients, it is indicated that the patients were clustered according to the datasets that were obtained rather than their phenotypic properties, suggesting that the model generated for TCIA may not accurately work for NCC without additional training.

The application workflow of our ML procedures is depicted (Figure 2). In order to generate an ML procedure for the ML model that is sufficiently accurate for use in multiple cohorts or datasets, we developed a novel pipeline, termed PABLO. The purpose of classification was to generate a discriminative model between GBM/LrGG and other mutation/epigenetics types.

In Figure 3, the blue bars represent classification performances that evaluate model accuracies for the TCIA dataset by five-fold cross validation in the absence of normalization or dimension reduction. When the various classifiers implemented in PABLO were simply applied to predict available pathological or genetic characteristics related to glioma, such as GBM/LrGG classification (GBM prediction) and IDH mutation prediction (IDH prediction), the AUROC of the method was 0.904 for GBM prediction and 0.867 for IDH prediction; scores were comparable to those of previous methods, including a study based on DL [13,45]. Because of the significant clinical importance and high comparability with the previous radiogenomic studies, we focused on GBM and IDH predictions hereafter.

On the other hand, a model that is trained for TCIA and shows the highest accuracy in cross validation does not always achieve a high accuracy on other cohorts. The grey and orange bars seen in Figure 3 indicate the accuracies obtained when the best models developed in TCIA were applied to the NCC validation and test set (0.383 and 0.392, respectively), showing a significant decrease in accuracy for both cases.

Next, we applied standardization of brain shape prior to image feature computation, which was used to improve the accuracy of the model for radiogenomics in multiple cohorts. The accuracies obtained when two spatial standardization platforms SPM and ANTs were applied are compared with those without standardization for GBM and IDH prediction (Figure 4A,B). As with the case without standardization methods, the accuracies obtained for GBM prediction within the TCIA dataset were higher than 0.9 with SPM and ANTs, while those obtained via NCC changed drastically, depending on the standardization method used. In particular, the accuracy obtained via the SPM standardization method remained higher than 0.80, indicating that standardization may enhance applicability of a model to multi-cohort analyses. The accuracies estimated for IDH prediction also showed a similar tendency to those for GBM prediction. As such, it is suggested that standardization may improve the generalization performance of the generated model.

Dimension reduction is another pre-processing procedure to extract or generate lower dimensions representing the differences between different classes, such as GBM/LrGG or IDH mutation, beyond cohort differences. For the purpose to examine the influence of dimension reduction, simple feature selection methods, PCA, and NMF with varying the dimension size were performed before the classification (Figure 4C,D). All accuracies obtained for both prediction targets were high for TCIA when using dimension reduction methods. However, the accuracies for NCC test and validation using PCA and NMF showed substantially higher performances than those via feature selection in GBM prediction. For IDH prediction, the accuracies of NMF were higher than those of the others. These results indicate that dimension reduction has the potential to reduce the cohort-specific systematic differences and increase the accuracy for NCC (even when the model was generated using the TCIA dataset only).

However, the model that achieved the highest accuracy for TCIA still did not achieve the highest accuracy for the NCC dataset, even after classification and dimension reduction were applied. Thus, we next assume a practical situation where a part of the local dataset, or NCC validation, is available to select a robust model among the generated models for TCIA. The accuracies of the two models are compared for the TCIA, NCC validation, and NCC test set to show which model is the best for the NCC test set (Figure 5). In either case, the model selected by the TCIA result was not the best, but worse for the NCC dataset, while the best model for the NCC validation set produced comparable accuracies for the TCIA dataset. As such, selecting the most robust model among all generated models according to their performances for partial validation data only is suggested to be an efficient way to obtain comparable prediction accuracy, even though the models are not re-trained for the new dataset.

We further tested whether the accuracies showed similar tendencies even after clinical information was integrated into the radiomic features. We combined image information with clinical information, such as sex, age, and KPS, for classification, and examined the changes in accuracy in the same manner shown in Figure 3, Figure 4 and Figure 5. Accuracies that were obtained using different standardization methods for GBM and IDH prediction are shown (Figure 6A,B). First, the accuracies obtained via TCIA and NCC overall tend to be higher than those obtained using only image information, indicating the significant contribution made by clinical information. In addition, the accuracies were less sensitive for the difference of the standardization methods used. Similar calculations performed for dimension reduction methods are shown (Figure 6C,D). These results indicated that while combining image features with clinical information substantially increased prediction performance, dimension reduction was not sufficiently effective in bringing about improvement as much as that shown in classification based on image features alone.

However, the importance of standardization and dimension reduction was revealed when the accuracies were compared between the models, showing the highest accuracy for the TCIA and NCC validation set (Figure 5 and Figure 7). As a result, the selected models that showed the best accuracy for the NCC validation dataset were based on both SPM for standardization and NMF for dimension reduction. This result suggests that a combination of standardization and dimension reduction may enhance the stability and reproducibility of predictions based on image features and clinical information across the different cohorts, such as the public training dataset and test set from local hospitals.

## 4. Discussion

Radiogenomics is a rapidly developing field of research to predict the genetic backgrounds of neoplasms, including gliomas using medical images including computed tomography, MRI, and ultrasound, among others [51,52]. Precise diagnoses for gliomas require surgically obtained specimens for further investigation of immunohistochemistry, fluorescence in situ hybridization, and DNA sequencing. Although it is reported that the extent of resection (particularly when close to complete resection) is associated with better prognoses in GBM [53], the extent of resection was not prognostic for IDH 1/2 mutant and 1p/19q co-deleted tumors, which are known to be sensitive to chemotherapy as well as radiotherapy [54,55]. Hence, the potential impact of radiogenomics is that it offers a non-invasive method to predict these genetic markers important for diagnosis and prognosis. In fact, the WHO classification of tumors will be updated soon, in 2021, and include the need of analysis for chromosome 7 and loss of whole chromosome 10, and TERT promoter mutation and EGFR amplification for the diagnosis of glioblastoma even though the tumor has histologically low grade glioma features. Therefore, the importance of genetic background detection is increasing along with the discovery of more genes involved in modulating the tumor growth. Recent studies revealed the heterogeneity of genetic background within the tumor, which is expected to be decoded by radiogenomic studies [56].

The objective of this study was to discover important factors to establish a classifier that is able to perform better, not only in the context of single-cohort cross validation, but also when the model is used in other hospitals. Due to requirements associated with maintaining confidentiality regarding patient images and clinical information, we assumed a general situation, where the classifier was required to be applicable to a local dataset, using parameters trained only by a public dataset. Classification accuracies obtained via cross validation in TCIA were notably higher than those obtained by applying the model to either of the NCC validation or test sets, indicating a loss of generality in the model generated and selected from the TCIA dataset (Figure 6 and Figure 8). However, the findings also showed the presence of models having accuracies that were similar to those of the best models, demonstrating that pre-processing via brain-embedding yielded a much more robust model over cohorts/hospitals and clinical information was as important as image data. Thus standardization and dimension reduction, particularly for the combination of SPM and NMF, have potential to generate models that can be used over multiple hospitals.

In addition to investigating different pre-processing and dimension reduction procedures, we also analyzed different brightness standardization methods and determined that changes in brightness did not completely fill the gap between TCIA and NCC images. Furthermore, the effect of ROI information was investigated and the results indicated that using tumor ROI did not improve prediction accuracies for IDH and MGMT mutations in GBM patients (Appendix A). Prediction accuracies obtained without ROI information showed different tendencies for TCIA and NCC datasets. While accuracy obtained via cross validation was higher than 0.8 for GBM and IDH prediction in TCIA, the highest accuracy models obtained via the NCC dataset were only 0.4-0.6 AUROCs, regardless of the pre-processing methods used (Appendix A). This suggested the importance of using edema ROI information based on classical radiomic features for prediction improvement across cohorts. The classification performances of each single feature are also varied, depending on the source data types. Hence, the classification performance of each feature may be substantially changed according to the input and ROI (Appendix A).

Furthermore, we examined the differences in accuracy associated with different ML classifiers and determined that the higher the accuracy of the model trained and evaluated on the same dataset was, the lower the accuracy of the model tested on the different dataset (Appendix A). In addition, the procedures without dimension reduction methods tended to yield a lower accuracy for cross-dataset applications instead of a higher accuracy via cross validation, compared to the ones with dimension reduction methods, indicating the overfitting potential of each ML classifier in the cases without dimension reduction.

## 5. Conclusions

In this study, we performed a comprehensive assessment of radiogenomic classification models using two glioma datasets that are suffering from systematic biases for hospitals involved in the dataset acquisition and image pre-processing. Our results suggest that the ML method that yielded the highest accuracy in a single dataset was likely to be overfitted and showed the severe decrease of prediction accuracies for another dataset. We further tested the impact of the implementation of standardization and dimension reduction procedures prior to classification. As a result, it enabled the development of ML methods that are less affected by the cohort differences. Our results advocate using caution in evaluating the accuracy of radiogenomic ML models in which the training and testing datasets are small or mixed, with a view to implement practical ML methods in radiogenomics.

## Figures and Tables

**Figure 1 cancers-13-03611-f001:**
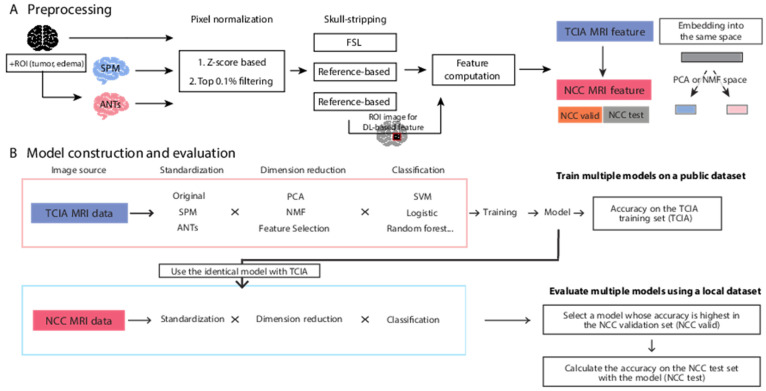
An overview of the dataset preprocessing and analytical methods used in this study. (**A**) Brain image data is applied to the multiple preprocessing steps; standardization of brain shape, pixel normalization, and skull-stripping. Then image features for radiogenomics are computed. To test the robustness of the trained models, we separated the whole NCC dataset to two independent partial NCC dataset: NCC validation and test set. The embedding space for dimension reduction is obtained from the TCIA dataset and applied to the partial NCC datasets independently. (**B**) In a machine learning workflow, all ML models were generated from TCIA data. The models were applied to TCIA (for cross validation), with half of the NCC data used for validation (termed NCC validation or NCC valid) and the other half of the NCC data (NCC test).

**Figure 2 cancers-13-03611-f002:**
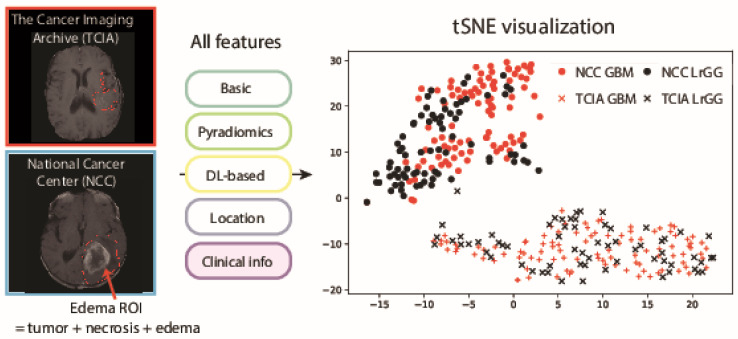
t-SNE visualization of all data from the TCIA and NCC datasets. Red and black colors depict GBM and LrGG patients, respectively, while crosses and circles indicate patients from TCIA and NCC datasets, respectively. The plots were based on image features calculated from MRI brain images and edema ROI annotation information and clinical features.

**Figure 3 cancers-13-03611-f003:**
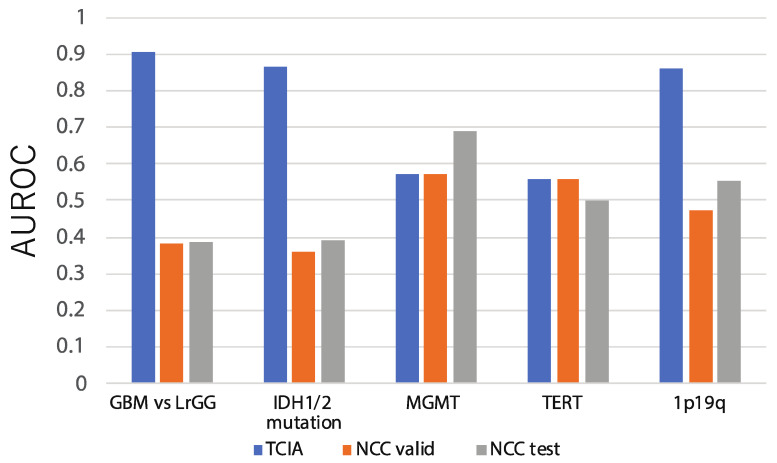
AUROCs were used as criteria to assess prediction accuracies of our machine learning workflow, without pre-processing, for 5 prediction problems: GBM/LrGG classification, IDH mutation existence, MGMT methylation status prediction, TERT promoter methylation prediction, and chr 1p19q co-deletion prediction. The accuracy of cross-validation results from TCIA (depicted as TCIA in Figure 2), as well as the accuracy of the application of the model to the NCC validation set (NCC valid) and the NCC test set (NCC test), are indicated by blue, orange, and grey bars, respectively.

**Figure 4 cancers-13-03611-f004:**
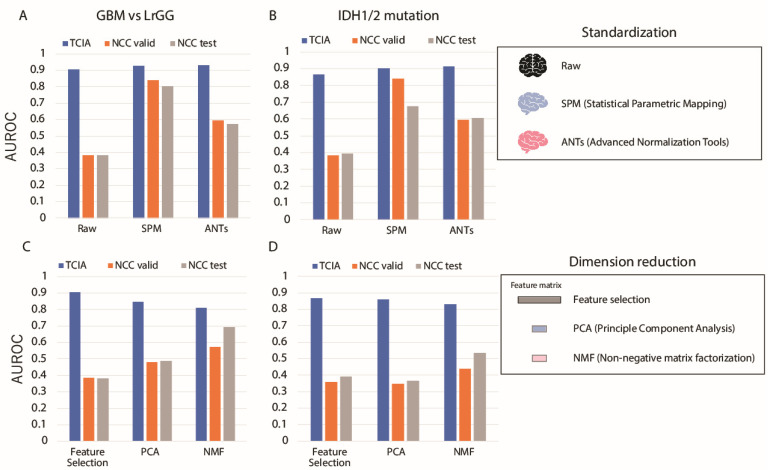
Changes in the classification performance of GBM (**A**,**C**) and IDH (**B**,**D**) prediction when standardization or dimension reduction methods were changed. (**A**,**B**), Changes in the accuracy of GBM and IDH prediction due to differences in the standardization method. Blue, orange, and gray bars correspond to TCIA cross validation, NCC validation, and NCC test accuracies, respectively. (**C**,**D**), Changes in the accuracy of GBM and IDH predictions due to differences in the dimension reduction method.

**Figure 5 cancers-13-03611-f005:**
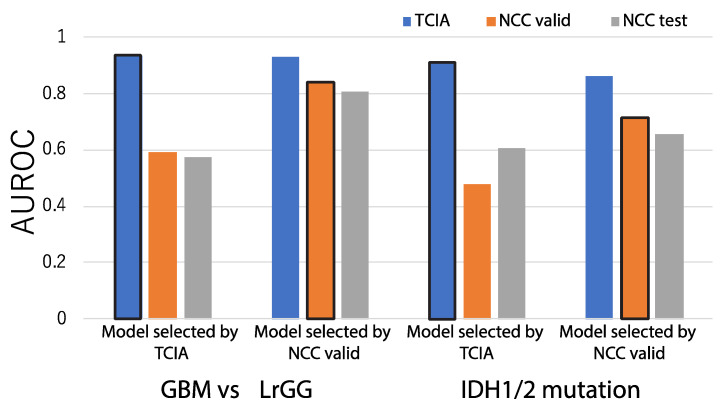
Comparison of classification performances between the highest accuracy model in the TCIA and NCC validation dataset. The leftmost chart shows accuracies yielded by the model that showed the highest accuracy under cross validation within TCIA. The second leftmost chart shows the accuracies yielded by the model that showed the highest accuracy via the NCC validation set. The two graphs on the right depict the same settings as the previous examples, except that the prediction target is IDH mutation prediction.

**Figure 6 cancers-13-03611-f006:**
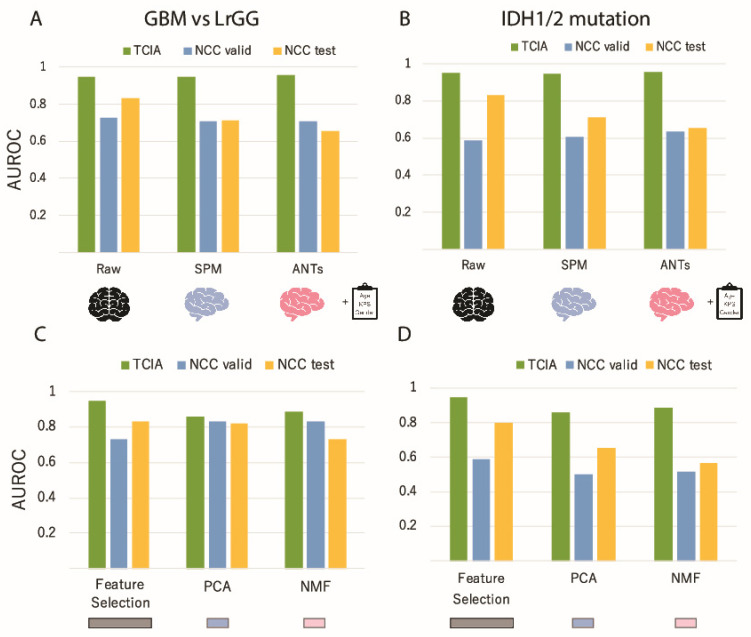
Changes in classification performance for GBM (**A**,**C**) and IDH (**B**,**D**) prediction due to differences between standardization and dimension reduction methods. The difference of this figure from Figure 4 is that input data comprised images as well as clinical information, whereas in Figure 4 only image features are used. (**A**,**B**) Comparison of accuracies due to differences in the standardization method for GBM and IDH prediction, respectively. (**C**,**D**), Comparison of accuracies due to differences in dimension reduction methods for GBM and IDH prediction, respectively.

**Figure 7 cancers-13-03611-f007:**
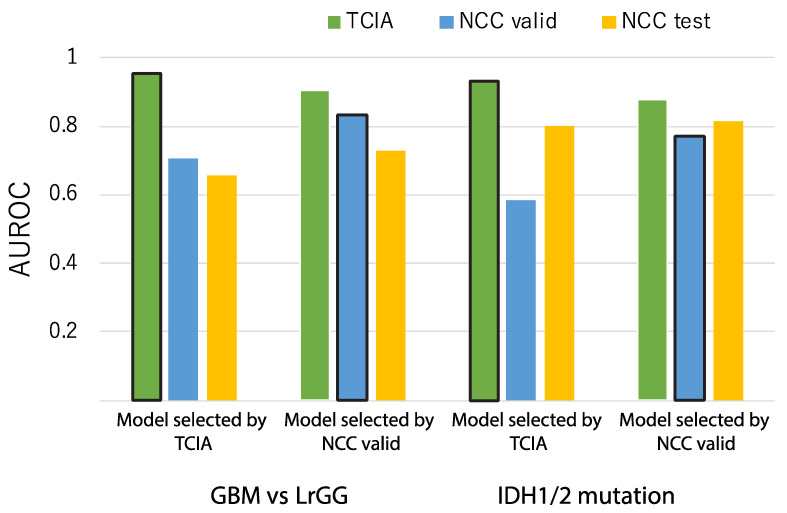
Comparison of classification performance between the highest accuracy models in TCIA and NCC validation. The experimental conditions were similar to those represented by Figure 5, except for the fact that input data contained clinical information. The two left and right graphs depict GBM and IDH prediction, respectively. In each group of 2 graphs, the left graph represents the model with the highest accuracy in TCIA while the right represents the model with the highest accuracy in the NCC validation set.

**Figure 8 cancers-13-03611-f008:**
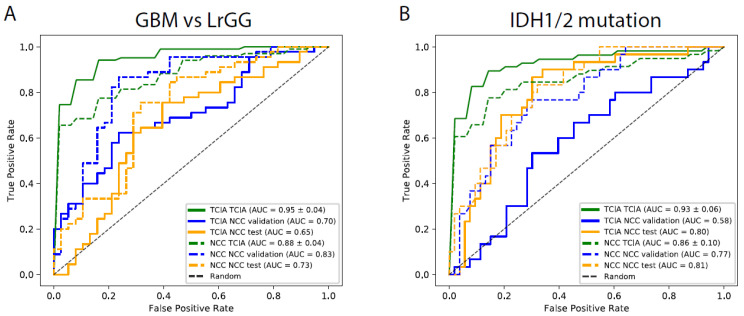
Receiver operating characteristic (ROC) curves for classification results. The ROC curves enable the visualization of results in Figure 7 for GBM (**A**) and IDH prediction (**B**), respectively. Solid and dotted lines show the curves for the highest accuracy model in TCIA and NCC validation set, respectively. Each color corresponds to that used in Figure 7, where green, blue, and yellow depict TCIA, NCC validation, and NCC tests, respectively.

**Table 1 cancers-13-03611-t001:** Demography of glioma patients in two different datasets used in this study, The Cancer Image Archive (TCIA) and National Cancer Center Hospital Japan (NCC). Each row represents the statistics or number of people assigned to each category for GBM and lower grade glioma patients. For genomic alterations, the number of people carrying IDH mutations, TERT promoter mutations, chr 1p19q codeletions, and higher levels of MGMT promoter methylation was separated from those of others via slash.

	TCIA		NCC	
	GBM	LGG	GBM	LwGG
Total	102	65	90	76
Age (average)	57.5	44.0	61.9	45.5
Gender (F/M)	32/55	38/27	39/51	29/47
KPS (average)	80.4	90.0	76.6	86.7
IDH mut vs wt	4/66	53/11	6/84	52/22
TERT mut vs wt	3/1	21/40	55/35	33/43
chr1p19q codel	0/83	13/52	2/48	25/50
MGMT met H vs L	20/29	52/13	34/56	46/30

The data provided by Arita et al., which consisted of IDH1/2 mutations, codeletion of chromosome 1p and 19q (1p19q codeletion), TERT promoter mutations, and MGMT methylation, were used to represent the genomic and epigenomic features of the NCC dataset [7]. All values were binarized according to a previous study.

## Data Availability

Segmentation labels and radiomic features for the pre-operative scans of the TCGA-LGG and -GBM collection can be downloaded from the TCIA database: https://www.cancerimagingarchive.net/ (accessed on 3 June 2021). Information on the biomarkers and demography of the patients were obtained from corresponding [40]. The statistics of the subjects in the NCC dataset were published in the previous study [7], whereas the NCC image dataset is not publicly available.

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
