# Peer review of "Assessing Versatile Machine Learning Models for Glioma Radiogenomic Studies across Hospitals"

_cancers, 2021, doi:10.3390/cancers13143611_

Round 1

Reviewer 1 Report

This is a study investigating the generalizability of radiomics-based models built on one dataset to a dataset at a different institution. A publicly available TCIA dataset (n = 159) was used to build a model and a local dataset (n = 166) was used to select the optimal model and test the accuracy of the model without re-training. The authors have introduced a standardization approach before testing the model which was shown to improve accuracy. Although this is an interesting and important area for validation of radiomics based models, there are significant flaws within the methodology as outlined below. Similarly, the figures are hard to read, the manuscript is hard to follow at times, and there are several discrepancies throughout.

Major Comments:

  1. Introduction - Page 2, line 79: The unmet need is not clear from the introduction. It is clear how the genetic information provides clinical utility for these patients. However, if you are getting this genetic information already then why do we need to get this information from imaging? The authors need to provide a well cited argument stating the clinical need and impact for this research question, such as:
    1. Why genetic information is not currently available at the time of imaging. For example, when and how is this usually taken in the clinical workflow?
    2. Do all patients currently get genetic testing? What are the limitations?
  2. Introduction – Page 2, line 89: Several claims in the introduction are lacking references. For example, “On the other hand, studies using local datasets showed significant diversity in prediction accuracy for the same prediction target.” Please provide references to back up this claim and other statements presented later in this paragraph.
  3. Introduction: The authors state the objective of this study was to determine if “the trained model is still effective for datasets from local hospitals.” However, this study only assessed a single hospital dataset, and the results cannot be generalized to more hospitals/dataset. The same goes for comments in the discussion/conclusion.
  4. Supplementary: The supplementary figures and tables were not available. The table and figure captions were provided in the Word document however the actual figures and tables were not available to this reviewer. Similarly, I do not see any information in the supplementary material for the ROI extraction (as is referred to in the text).
  5. Methods: Please provide specific details on the scanners and scan protocols for all patients analyzed in this study.
  6. Methods – ROI Segmentation: Another limitation of this study is the single user and manual segmentation of the ROIs. It is well known that variability both between and within observers can significantly impact feature extraction and subsequent model building. Therefore, the authors need to show stability of the radiomic features given differences in segmentations by different users (or through segmentation perturbations).
  7. Methods – Classification: Based on the methods, the abstract and introduction does not reflect what was actually done.
    1. The methods state the model was applied to half the NCC data. Why wasn’t the full NCC data used to test the TCIA model?
    2. Similarly, 5-fold cross validation was used to fit the model, and this would result in 5 models. How was the optimal model determined? Now seeing Figure 1C, I think half the NCC data was used to select the optimal model? However, this results in information leakage between the two datasets, as the second data set was used to determine the optimal model. Therefore, the validity of this study is questionable as the results are going to be overly optimistic. This is evident in the results presented in Figure 3. Without the “information leakage” you are only seeing AUCs of 0.5-0.6 on the NCC datasets.
  8. Methods: Overall the authors have investigated several classifiers and feature selection approaches. This results in multiple testing of the radiomic features by using several different methods. Therefore, the appropriate statistical analysis needs to be performed. Developing all of these models and then using a separate dataset to “select the best one” is not appropriate. This will increase the risk of a type 1 error.
  9. Figure 1: This figure is extremely hard to read. Figure 1A should be a separate Table and all other subfigures should be individual figures. Similarly, Figure 3C only has 3 classifiers but six are listed? Similarly, all plot axes need to be labelled.
  10. Results: Overall the results are hard to follow. Headings to describe each different experiment would help with the flow.
    1. The authors have performed five different classifications and then randomly focus down on 2 for simplicity. Why wasn't the full study performed on only two classifications?
    2. Several methods are described in the results including introduction of the PABLO pipeline.
    3. It is unclear how dimensionality reduction would impact the NCC datasets. Was the model not locked in the TCIA cohort prior to validation? The authors state that “dimension reduction has a potential to reduce the cohort specific systematic differences and increased the accuracy for NCC.” It is well known the smaller number of features you use the more generalizable a model will be. It is unclear the need for this analysis? Aren't these results just showing that different feature selection approaches impact the model, which is well known?

Minor Comments:

  1. Introduction - Page 2, line 68: The five-year survival rate of GBM is presented, however for comparison it would be useful to show similar data for low grade gliomas.
  2. Introduction – Page 2, lines 92-100: This paragraph is very hard to follow. Please re-write this paragraph with more precision, for example please clarify what you mean by the following:
    1. “sample size is not enough to obtain quantitative results”
    2. “stability of the best methods”
    3. “a spatial standardization”
  3. Introduction – Page 3, lines 104: This is not the correct definition of transfer learning, which actually refers to when a model developed for a task is reused as the starting point for a model on a second task. This is not the same as validating a pre-existing model on an external dataset, which is what the authors are describing.
  4. Methods: – ROI Segmentation: “Pixels annotated as “necrosis” or “enhancing tumor region” were labelled as tumor, and regions annotated as “necrosis,” “edema”, “non-enhancing tumor” and “enhancing tumor region” were treated as edema ROI.” How can the same annotations be both tumor and edema? Were all tumor regions also defined as edema? If so, this isn’t clear in the NCC dataset. A figure showing the ROIs would be beneficial.
  5. Methods – Preprocessing: This section is a bit hard to follow on what exactly was done on each dataset. A flow diagram would be beneficial to show each step applied to each dataset.
  6. Methods – Feature Extraction: The GitHub link does not have any details on feature extraction. Similarly, it is unclear why basic statistics are measured in addition to the Pyradiomics features. Given that all of these features are contained within Pyradiomics. Finally, it was stated that “features were computed only for edema (tumor, or all brain) regions.” Why is tumor and brain in brackets? It is very hard to follow what regions were used for feature extraction. Why segment both if you’re only using one?

Author Response

Response to Reviewer #1:

Dear Professor,

We are grateful for your consideration of our manuscript entitled “Assessing Versatile Machine Learning Models for Glioma Radiogenomic Studies across Hospitals” (manuscript ID: cancers-1181199) by Kawaguchi et al. and appreciate your helpful comments.

Replies to the comments are as follows:

Major Comments:

Comment 1: Introduction - Page 2, line 79: The unmet need is not clear from the introduction. It is clear how the genetic information provides clinical utility for these patients. However, if you are getting this genetic information already then why do we need to get this information from imaging? The authors need to provide a well cited argument stating the clinical need and impact for this research question, such as:

  1. Why genetic information is not currently available at the time of imaging. For example, when and how is this usually taken in the clinical workflow?
  2. Do all patients currently get genetic testing? What are the limitations?

Reply: We appreciate the reviewer’s attentive comment about the motivation of our research. The advantage of radiogenomic studies for glioma is that the information of MRI is available preoperatively. WHO grading system for glioma is changed recently so that genetic analysis is required for the precise diagnosis. However, because the actual analysis of genetic background requires tumor tissues taken from resection or biopsy of the tumor and it takes time for sequencing, the radiogenomic prediction has a potential to support the early determination of the appropriate treatment in a non-invasive way. Another point related with your comment 1-1 and 1-2 is the genetic heterogeneity of the glioma tissue, which is found by single cell RNA-seq. Due to the deep relationship between the phenotypic trait and appearance of tumor in MRI, the image-based analysis might capture the locational difference of tumor cell progression from image texture difference. We added the description below in Introduction, Page 2, Line 81.

“In addition, recently WHO grading system of central nervous system tumors has been updated to require genetic testing including IDH mutation, TERT promoter mutation, H3K27M mutation, 1p/19q codeletion for the precise diagnosis of gliomas. Currently these amendments can offer better understanding of glioma biology, but also raise a new problem that all the tumor tissues supposed to be a glioma should be ideally taken by surgery or biopsy and analyzed for genetic testing, but not all the countries have enough pathological diagnosis capacity.”

Comment 2: Introduction – Page 2, line 89: Several claims in the introduction are lacking references. For example, “On the other hand, studies using local datasets showed significant diversity in prediction accuracy for the same prediction target.” Please provide references to back up this claim and other statements presented later in this paragraph.

Reply: Thank you for your comments. We added descriptions and references, starting from Page 2, Line 99 as below and added references of [28, 35] to the following sentence.

“For example, radiogenomic features are found to predict MGMT methylation status with 83% accuracy using deep learning [28] although another study achieved the accuracy around 62% for the same data source [35]. …”

Comment 3: Introduction: The authors state the objective of this study was to determine if “the trained model is still effective for datasets from local hospitals.” However, this study only assessed a single hospital dataset, and the results cannot be generalized to more hospitals/dataset. The same goes for comments in the discussion/conclusion.

Reply: We appreciate the reviewer for the keen comment. The reviewer's comments are quite important and our aim in this study is to see how robustly the classifier trained on public data works in local hospitals. A larger dataset is needed to investigate how generalizable the results are, but due to the rarity of gliomas, it is not easy to expand the local data. However, as recently published (ref 35, Takahashi, et al. Cancers. 2021 Mar 19;13(6):1415. doi: 10.3390/cancers13061415), we have already collected larger dataset and this point is the subject of next project. In addition, the TCIA data is a mixture of data from multiple hospitals and pre-processed data. Therefore, we focused on the comparison between anonymized pre-processed open data from different resources of TCIA and real-world data taken at local hospital. Moreover, our strategy in which a robust model is selected using a small part of local dataset from a variety of trained models can be a platform to evaluate the systematic differences between the datasets as well as the classifier transferability. We believe that our study is able to suggest the problem and potential solution sufficiently and will be the basis for future investigation. We have added the description below to Page 3, Line 130.

“According to our result, the best model is shown to suffer from the over-fitting problem for another dataset, especially when the model does not include either standardization or dimension reduction processes. It would be a robust and practical solution to choose the better model from multiple models trained for the public dataset using partially annotated local dataset.“.

Comment 4: Supplementary: The supplementary figures and tables were not available. The table and figure captions were provided in the Word document however the actual figures and tables were not available to this reviewer. Similarly, I do not see any information in the supplementary material for the ROI extraction (as is referred to in the text).

Reply: Thank you for pointing out the problem. We upload Supplementary Tables and Figures and correct the misplacement of the citation to Supplementary materials.

Comment 5: Methods: Please provide specific details on the scanners and scan protocols for all patients analyzed in this study.

Reply: Thank you for your comment. We added Supplementary Table 2 to show the details of imaging protocols for all patients analyzed in this study.

Comment 6: Methods – ROI Segmentation: Another limitation of this study is the single user and manual segmentation of the ROIs. It is well known that variability both between and within observers can significantly impact feature extraction and subsequent model building. Therefore, the authors need to show stability of the radiomic features given differences in segmentations by different users (or through segmentation perturbations).

Reply: As you pointed out, we recognize that the dependence of manual annotation perturbations is a limitation of the study using manual segmentation. In our paper, the annotations are performed by different radiologists within TCIA and between TCIA and NCC datasets. Since the trained classifiers are expected to be robust for the difference of manual annotation, the segmentation perturbation issues are not completely but partly considered. One of the solutions to avoid the perturbation is the automatic segmentation, but we thought that it is out of scope in this paper.

Comment 7: Methods – Classification: Based on the methods, the abstract and introduction does not reflect what was actually done.

  1. The methods state the model was applied to half the NCC data. Why wasn’t the full NCC data used to test the TCIA model?
  2. Similarly, 5-fold cross validation was used to fit the model, and this would result in 5 models. How was the optimal model determined? Now seeing Figure 1C, I think half the NCC data was used to select the optimal model? However, this results in information leakage between the two datasets, as the second data set was used to determine the optimal model. Therefore, the validity of this study is questionable as the results are going to be overly optimistic. This is evident in the results presented in Figure 3. Without the “information leakage” you are only seeing AUCs of 0.5-0.6 on the NCC datasets.

Reply: Thank you for your comments. Our replies to the comments are as below:

7-1. The models were trained on the TCIA data. The optimal model was selected with half of the NCC data (validation data), and evaluated on the rest of the NCC data (test data). We modified Figure 1 for better understanding.

7-2. After 5-fold cross validation, we took the average of the AUCs for training data. The models that are applied to NCC datasets were optimized for all training data independently. Since the test data is independent from the training and validation data, there is no information leakage between datasets. Less heterogeneity in the TCIA dataset might cause the optimistic results you pointed out.

Comment 8: Methods: Overall the authors have investigated several classifiers and feature selection approaches. This results in multiple testing of the radiomic features by using several different methods. Therefore, the appropriate statistical analysis needs to be performed. Developing all of these models and then using a separate dataset to “select the best one” is not appropriate. This will increase the risk of a type 1 error.

Reply: Thank you for your comment regarding statistical analysis. Yes, we showed that “the best one” is not appropriate especially for the same dataset, and also the model generated from TCIA data is overfitted to TCIA data, causing a type 1 error you mentioned. To the best of our knowledge, there is no statistically sound comparison of ROC curves in this case. Therefore, the significance of the AUROC distribution differences between the cross validation and inter-hospital validation is tested (Supplementary Figure 7). A violin plot shows the distribution of AUROCs is highly different depending on the type of machine learning classifier as well as the existence of systematic biases between training and test datasets. We detected the significant p-values by Wilcoxon’s signed rank test after Bonferroni correction (q-value < 0.05) for all 8 ML classifiers, suggesting the evident decrease of classification performances.

Comment 9: Figure 1: This figure is extremely hard to read. Figure 1A should be a separate Table and all other subfigures should be individual figures. Similarly, Figure 3C only has 3 classifiers but six are listed? Similarly, all plot axes need to be labelled.

Reply: Thank you for your supportive suggestions. We separated Figure 1 into new Figure 1-3 and Table 1. We are sorry for unclear figures. Classes we checked are five different classes shown in Figure 1D. GBM and IDH mutation classifications are shown in the main text, and the others are in the supplementary material. We hope our comments clarify your questions.

Comment 10: Results: Overall the results are hard to follow. Headings to describe each different experiment would help with the flow.

  1. The authors have performed five different classifications and then randomly focus down on 2 for simplicity. Why wasn't the full study performed on only two classifications?
  2. Several methods are described in the results including introduction of the PABLO pipeline.
  3. It is unclear how dimensionality reduction would impact the NCC datasets. Was the model not locked in the TCIA cohort prior to validation? The authors state that “dimension reduction has a potential to reduce the cohort specific systematic differences and increased the accuracy for NCC.” It is well known the smaller number of features you use the more generalizable a model will be. It is unclear the need for this analysis? Aren't these results just showing that different feature selection approaches impact the model, which is well known?

Reply: Thank you for your comments.

  1. From the viewpoint of the physicians, GBM/LrGG and IDH mutations are clinically important features because these differences largely affect the prognosis of the patients and the decision of the treatment strategy. Hence, we focused on GBM/LrGG and IDH mutation classifications. We emphasized the description in the manuscript, Page 8, Line 295.
  1. We moved the explanation of PABLO from the Results to the Methods.
  1. Yes, the model including the dimension reduction and parameters of classifiers is fixed depending on the TCIA dataset. The embedding of the feature matrix by PCA and NMF is learned via the TCIA cohort then applied to the NCC dataset. It would be stable if the small number of effective features is already known. However, the problem assumed in this study is that there is no information which feature can predict the genetic background efficiently. While the AUROC of each single feature may be used to extract the high performance features, the results of feature selection using logistic LASSO, which is corresponding to such strategy, suggest the possibility of strong overfitting for inter-hospital comparison. Hence, the application of dimension reduction is considered to be a suitable method that captures low dimensional space from a huge feature matrix. Additionally, we observed that not all classifiers necessarily show the best result with a smaller number of features after dimension reduction.

Minor Comments:

Comment 1: Introduction - Page 2, line 68: The five-year survival rate of GBM is presented, however for comparison it would be useful to show similar data for low grade gliomas.

Reply: Thank you for your comment. We added the information of five-year survival rate of glioblastoma and low grade glioma patients updated from the same reference at Page 2, Line 69.

Comment 2: Introduction – Page 2, lines 92-100: This paragraph is very hard to follow. Please re-write this paragraph with more precision, for example please clarify what you mean by the following:

  1. “sample size is not enough to obtain quantitative results”
  2. “stability of the best methods”
  3. “a spatial standardization”

Reply: We appreciate your comment. We rewrote the paragraph to clarify the points as below.

  1. Page 3, Line 106. “Where the number of patients……….
  2. Page 3, Line 107. “Furthermore, most studies examine the prediction accuracy stability of the best methods within one dataset and do not show the results from the viewpoint of the performance stability across the datasets, particularly for the difference between public anonymized dataset and local data [38]. The reproducibility of radiogenomic studies has already gathered attention for the evaluation of different preprocessing strategy and tumor annotation [39,40].”
  3. Line 114. “A spatial standardization, which is an approach to scale each brain image to a reference brain, is one promising way to reduce the impact of the systematic differences while even the choice of standardization is impactful for the performance stability [43].”

Comment 3: Introduction – Page 3, lines 104: This is not the correct definition of transfer learning, which actually refers to when a model developed for a task is reused as the starting point for a model on a second task. This is not the same as validating a pre-existing model on an external dataset, which is what the authors are describing.

Reply: Thank you for your comment. We acknowledged the reviewer’s comment. We refer to the case assumed in this study as one type of transfer learning “transductive transfer learning”, or domain adaptation as defined in Pan SJ and Yang Q (2009). However, it might be confusing to the readers about our experiment as pointed out by the reviewer. Thus, we changed the description in Line 120 as follows.

“This type of problem is a main subject of transfer learning in the ML domain because re-training of the entire models demands a huge resource as well as annotated training data from a local hospital [44].”

Comment 4: Methods: – ROI Segmentation: “Pixels annotated as “necrosis” or “enhancing tumor region” were labelled as tumor, and regions annotated as “necrosis,” “edema”, “non-enhancing tumor” and “enhancing tumor region” were treated as edema ROI.” How can the same annotations be both tumor and edema? Were all tumor regions also defined as edema? If so, this isn’t clear in the NCC dataset. A figure showing the ROIs would be beneficial.

Reply: Yes, the edema region includes the tumor region inside in our definition. The tumor and edema ROI represents the regions enhanced in Gd-T1WI and high intensity lesion in T2WI and FLAIR, respectively, plus their inside. We clarify our definition as follows.

Page 4, Line 222: Image features are computed for the region inside of the outline.

Page 4 , Line 227: which corresponds to the edema region and its inside.

Additionally, we added the simple description to Figure 2 with the examples of edema ROI.

We carried out the classification validation using no-edema, tumor, and edema ROI. Main manuscript only contains the results of edema, which shows the highest performance overall as expected, and others are included in the Supplementary Figure 1-5.

Comment 5: Methods – Preprocessing: This section is a bit hard to follow on what exactly was done on each dataset. A flow diagram would be beneficial to show each step applied to each dataset.

Reply: Thank you for your comment. We improved the flow diagram in Figure 1 for better understanding.

Comment 6: Methods – Feature Extraction: The GitHub link does not have any details on feature extraction. Similarly, it is unclear why basic statistics are measured in addition to the Pyradiomics features. Given that all of these features are contained within Pyradiomics. Finally, it was stated that “features were computed only for edema (tumor, or all brain) regions.” Why is tumor and brain in brackets? It is very hard to follow what regions were used for feature extraction. Why segment both if you’re only using one?

Reply:  We included additional feature information in the README at PABLO repository. We included some basic features such as pixel sizes and brightness distribution of annotated regions. While many are overlapped with pyradiomic features, some features, such as percentiles of the pixel brightness from 0 to 100 in 10% increments or the size of ROI, are only applied in basic features. These basic features are computed for not only the tumor and edema ROI but also whole brain regions as a control for patient specific biases. Supplementary Figure 6 shows the distribution of AUROCs of each type of features in NCC cross validation. As expected, the features from the whole brain do not show substantial contribution to improve the classification performances in general. We make it clear that the main manuscript contains the results of edema ROI only and others are included in Supplementary data at Page 6, Line 229 as follows.

“The results shown in the main script are obtained using edema ROIs because of the high prediction performances. Those with tumor ROIs and without ROIs are found in Supplementary Figure 1-5.”

Reviewer 2 Report

The authors describe machine learning approach to classification of radiogenomic images from distinct data sources while suggesting standardization and dimension reduction procedures to avoid overfitting and to increase the overall model accuracy.
The manuscript preparation seems to be sound. It is comprehensively written, based on a comparison of different data sources, and material has been made openly available for reproducibility.
Regarding figures, axes label could be added and abbreviations should be described within figure legends for better comprehensibility, especially in case of Figure 2.
I cannot suggest any additional necessary amendments, only future prospect on including additional data sources and maybe focusing details as pediatric differences or including additional genomic features for follow-up studies.
Worthwhile to mention would be the upcoming update of the WHO classification scheme on central nervous system tumors as [Rushing, E.J. WHO classification of tumors of the nervous system: preview of the upcoming 5th edition. memo (2021). https://doi.org/10.1007/s12254-021-00680-x].

Author Response

Response to Reviewer #2:

Dear Professor,

We are grateful for your consideration of our manuscript entitled “Assessing Versatile Machine Learning Models for Glioma Radiogenomic Studies across Hospitals” (manuscript ID: cancers-1181199) by Kawaguchi et al. and appreciate your helpful comments.

Reply to the comment is as follows:

Comment: The authors describe machine learning approach to classification of radiogenomic images from distinct data sources while suggesting standardization and dimension reduction procedures to avoid overfitting and to increase the overall model accuracy.

The manuscript preparation seems to be sound. It is comprehensively written, based on a comparison of different data sources, and material has been made openly available for reproducibility.

Regarding figures, axes label could be added and abbreviations should be described within figure legends for better comprehensibility, especially in case of Figure 2.

I cannot suggest any additional necessary amendments, only future prospect on including additional data sources and maybe focusing details as pediatric differences or including additional genomic features for follow-up studies.

Worthwhile to mention would be the upcoming update of the WHO classification scheme on central nervous system tumors as [Rushing, E.J. WHO classification of tumors of the nervous system: preview of the upcoming 5th edition. memo (2021). https://doi.org/10.1007/s12254-021-00680-x].

Reply: We appreciate your thoughtful comments. We updated the legends of Figure 2-5 to improve interpretability. Additionally, the description about the new WHO grading is added in Discussion, Page 12, Line 411 as below.

“In fact, WHO classification of tumors will be updated soon in 2021, which includes the need of analysis for chromosome 7 and loss of whole chromosome 10, TERT promoter mutation and EGFR amplification for the diagnosis of glioblastoma even though the tumor has histologically low grade glioma features. Therefore, the importance of genetic background detection is increasing along with the discovery of more genes involved in modulating the tumor growth. Recent studies revealed the heterogeneity of genetic back-ground within the tumor, which is expected be decoded by radiogenomic studies [56].”

Reviewer 3 Report

In the current study, the authors present radiogenomic machine learning models for glioblastoma using several cohorts from different organizations. The study is well organized, and the manuscript is well written. The study will motivate the readers to do this kinds of studies because the current results clearly indicate that the kind of studies can be drawn by integration of several studies each of which lacks statistical power due to the small numbers of analyzed cases.

Although the authors can establish the models to predict the genomics, it is welcomed to check whether the models can predict the clinical courses, if possible.

Author Response

Response to Reviewer #3:

Dear Professor,

We are grateful for your consideration of our manuscript entitled “Assessing Versatile Machine Learning Models for Glioma Radiogenomic Studies across Hospitals” (manuscript ID: cancers-1181199) by Kawaguchi et al. and appreciate your helpful comments.

Reply to the comment is as follows:

Comment: In the current study, the authors present radiogenomic machine learning models for glioblastoma using several cohorts from different organizations. The study is well organized, and the manuscript is well written. The study will motivate the readers to do this kinds of studies because the current results clearly indicate that the kind of studies can be drawn by integration of several studies each of which lacks statistical power due to the small numbers of analyzed cases.

Although the authors can establish the models to predict the genomics, it is welcomed to check whether the models can predict the clinical courses, if possible.

Reply: Thank you very much for your thoughtful comments. Indeed, the prediction of the clinical courses of the patients with glioma is the subject of our next challenge. For example, the prediction of the response to chemotherapy or radiotherapy, the diagnosis whether recurrence or radiation necrosis at the enlargement of MRI lesions will be investigated in the future.